# Evaluation of Mechanical and Permeability Characteristics of Microfiber-Reinforced Recycled Aggregate Concrete with Different Potential Waste Mineral Admixtures

**DOI:** 10.3390/ma14205933

**Published:** 2021-10-09

**Authors:** Rayed Alyousef, Babar Ali, Ahmed Mohammed, Rawaz Kurda, Hisham Alabduljabbar, Sobia Riaz

**Affiliations:** 1Department of Civil Engineering, College of Engineering, Prince Sattam Bin Abdulaziz University, Alkharj 16273, Saudi Arabia; r.alyousef@psau.edu.sa (R.A.); h.Alabduljabbar@psau.edu.sa (H.A.); 2Department of Civil Engineering, COMSATS University Islamabad-Sahiwal Campus, Sahiwal 57000, Pakistan; babar.ali@cuisahiwal.edu.pk; 3Civil Engineering Department, College of Engineering, University of Sulaimani, Sulaymaniyah 46001, Iraq; ahmed.mohammed@univsul.edu.iq; 4Department of Technical Highway Engineering, Technical Engineering College, Erbil Polytechnic University, Erbil 44001, Iraq; 5Scientific Research and Development Center, Nawroz University, Duhok 42001, Iraq; 6CERIS, Civil Engineering, Architecture and Georresources Department, Instituto Superior Técnico, Technical University of Lisbon, Av. Rovisco Pais, 1049-001 Lisbon, Portugal; 7Department of Civil Engineering, University College of Engineering and Technology, Bahauddin Zakariya University, Multan 66000, Pakistan; sobiariaz@bzu.edu.pk

**Keywords:** mineral admixtures, microfiber, wastes, recycled aggregates, mechanical performance, chloride durability

## Abstract

Plain recycled aggregate concrete (RAC) struggles with issues of inferior mechanical strength and durability compared to equivalent natural aggregate concrete (NAC). The durability issues of RAC can be resolved by using mineral admixtures. In addition, the tensile strength deficiency of RAC can be supplemented with fiber reinforcement. In this study, the performance of RAC was evaluated with individual and combined incorporation of microfibers (i.e., glass fibers) and various potential waste mineral admixtures (steel slag, coal fly ash (class F), rice husk ash, and microsilica). The performance of RAC mixtures with fibers and minerals was appraised based on the results of mechanical and permeability-related durability properties. The results showed that generally, all mineral admixtures improved the efficiency of the microfibers in enhancing the mechanical performance of RAC. Notably, synergistic effects were observed in the splitting tensile and flexural strength of RAC due to the combined action of mineral admixtures and fibers. Microsilica and rice husk ash showed superior performance compared to other minerals in the mechanical properties of fiber-reinforced RAC, whereas slag and fly ash incorporation showed superior performance compared to silica fume and husk ash in the workability and chloride penetration resistance of RAC. The combined incorporation of microsilica and glass fibers can produce RAC that is notably stronger and more durable than conventional NAC.

## 1. Introduction

There are many aspects of the construction sector that are responsible for several of the ongoing environmental and sustainability issues. These aspects include, but are not limited to, mining of raw materials for the aggregate and cement industries, manufacturing of cement, and hauling of cement and aggregates to concrete plants or construction projects. To preserve the present condition of the environment and to tackle climate change challenges, aggregate and cement industries are being monitored by environmental protection agencies to reduce their demands for natural resources.

According to the Freedonia Group, the worldwide annual demand for construction aggregates is forecast to increase at the rate of 2.3% to 47 billion metric tons in 2023 [1]. The negative impacts of aggregate mining on the environment include the destruction of natural habitation, loss of fertile land, air contamination, erosion, and changes in the scenery of the mining area [2]. On the other hand, because of increased urbanization, the removal of old and expired infrastructure has also vastly increased the quantities of CDW. Negative effects of CDW disposal include loss of valuable and fertile lands to solid wastes, a reduction in the capacity of landfills, and the requirement of efficient solid waste management systems. Toxicity levels of grounds can also increase due to DW disposal [3]. In 2016, China produced about 15 billion metric tons of CDW, which created many social and environmental issues [4]. The CDW production rate of the European Union is about 350 million metric tonnes per annum [5]. A large fraction of CDW can easily be recycled to manufacture aggregates for the construction industry. By using RA to manufacture concrete, quantities of CDW can be efficiently reduced. Sustainability issues can also be addressed to some extent by using RAC [6,7,8] at the price of little reduction in engineering performance [9,10,11,12,13]. The applications of RAC can decrease the carbon emissions related to concrete manufacturing [9,14,15].

There are some drawbacks to using RAC that should be addressed to increase its use in the construction industry. RAC shows lower mechanical and durability performance compared to natural aggregate (NA) concrete (NAC). High porosity is the key cause of RAC’s [12,16,17] poor strength and durability compared to NAC. Studies have shown that the properties of RAC can be modified and improved by utilizing additional materials, e.g., mineral and chemical admixtures [9,11,17,18,19,20,21,22,23,24,25,26], and fiber [12,22,25,27,28,29,30] surface treatment methods [31,32,33]. By adjusting the mix design [34,35,36] using the equivalent mortar volume method, RAC with properties comparable to those of the NAC can be produced. Among all techniques, the utilization of mineral admixtures, e.g., MS, FA, blast furnace steel slag (BFS), and RHA, to improvise the properties of RAC is an economical and eco-friendly approach.

MS (ultrafine and reactive powder of silica) is accumulated as a waste product of silicon and ferro-silicon production. MS has a very high content of amorphous silica, which can effectively consume free lime due to a quick pozzolanic reaction. The properties of RAC with MS have been reported in the literature. Dilbas et al. [21] showed that RAC with 30–40% RCA and 5% MS exhibited compressive strength almost equal to that of Portland cement NAC (without no MS). With 10% MS, RAC incorporating 30–40% RA showed noticeably higher compressive strength than that of the Portland cement NAC. Kou et al. [18] revealed that RAC prepared with 10% MS and 50% RA showed comparable properties to those of conventional NAC. Kou et al. [18] also showed that among FA, BFS, and metakaolin, MS was superior in enhancing the compressive and tensile performance of RAC. 

BFS is obtained by pulverizing a stone-like waste matter separated from the smelting process of iron ores. It is rich in silica, lime, and alumina, and it also possesses pozzolanic properties. BFS has been proved useful in the durability of RAC. A study [37] reported that concrete incorporating 25–50% RA and 10% BFS exhibited a higher compressive strength and lower permeability compared to conventional NAC. Anastasiou et al. [38] reported that electric arc furnace slag recovers some strength loss and improves the durability of RAC by strengthening the bond between the binder matrix and RA. Kou et al. [18] reported that 55% of BFS incorporation decreased the mechanical performance of RAC substantially compared to that of the Portland cement NAC. Despite the loss in mechanical strength, RAC with 50% BFS and 100% RA content showed better chloride ion penetration resistance than that of NAC. 

FA is produced as an aluminosilicate powdered waste of coal-fed electricity generation plants. FA is rich in silica and alumina and acts as a pozzolanic material. Cement replacement with FA improves the ultimate strength of concrete, but it can drastically increase the chloride ion penetration resistance of concrete [17,26]. RAC with different levels of FA has been studied widely considering the mechanical performance of the material. FA also improves the workability of RAC and reduces the requirement of water-reducing chemicals to attain the required level of flow in fresh concrete [16]. Poon et al. [17] reported that the drawbacks of RAC (such as poor durability) can be minimized by the utilization of 25–35% FA as a cement substitution material. 

RHA is produced as a by-product by burning silica-rich rice grain shells for energy. It is a highly reactive, silica-rich pozzolanic material when prepared from the calcination of rice hulls (protective coverings of rice grains) well below 780 °C. The authors of [39] showed the properties of concrete with various replacement levels of cement with RHA. They also reported that RHA did not show any effect on compressive strength at an early age, but it improved the strength at later ages. The authors of [40] investigated the performance of RAC with 20% RHA. They demonstrated that RHA-containing concrete mixes showed superior mechanical performance than that of OPC-based mixes. RHA can improve the overall mechanical and durability performance of RAC [40]. Tangchirapat et al. [41] stated that using 20 to 35% RHA in RAC can provide a higher compressive strength than that of plain RAC.

Demands for sustainable, ductile, and high-strength composites (with high tensile strength) have increased globally over recent years. PRAC, despite being sustainable and environmentally friendly, is susceptible to cracking under tensile/flexural loadings (leading to a brittle failure). PRAC may also crack due to plastic and drying shrinkage of concrete. Discrete fiber reinforcement is a useful means to control the formation and proliferation of cracks in concrete that consequently improves the tensile strength [42,43], flexural strength [44,45], and energy absorption capacity [46,47] of cement-based composites. FRRAC, similar to fiber-reinforced NAC, experiences the dispersion issues of fibers. Microfibers such as glass and polypropylene filaments tend to accumulate in a fresh mix at higher doses [48,49]; therefore, fibers tend to induce air voids and lead to underutilization of fiber reinforcement. The introduction of air voids is highly damaging to the compressive properties of concrete [12]. Fibers are also damaging to the economy of concrete due to their higher cost and the requirement of water-reducing admixtures to achieve the desired fluidity [50].

The production of FRRAC with waste mineral admixtures can help in overcoming the issues that occur with the use of high fiber doses. Wu et al. [51] showed that MS helps in enhancing the efficiency and dispersion of steel fiber by increasing the bond strength between the fibers and binder. Admixtures such as FA and BFS are useful in improving the workability due to their lubricating effects, and these admixtures have also shown water-reducing effects on the fresh properties of concrete [9,18]. Therefore, FA and BFS can overwhelm the adverse effects of fibers on the flowability of FRRAC. Furthermore, the replacement of some parts of cement with a mineral admixture substantially reduces the carbon footprint of FRRAC. The reduction level of carbon footprint is related to the type of mineral admixture and replacement level of cement. The development of these types of composites can resolve the major issues associated with conventional plain cement concretes.

## 2. Research Significance

To the authors’ best knowledge, very few studies have investigated the properties of RAC made with both mineral admixtures (MS [22] and BFS [52]) and fiber reinforcement. Notably, there is no information on the properties of glass fiber-reinforced RAC (GFRRAC) with cheaply available and sustainable sources of mineral admixtures, e.g., RHA, MS, and BFS. Ali et al. [53,54] studied the performance of GFRRAC with FA and MS and reported that mineral admixtures can improve the bond strength of glass fibers. FA also minimized the negative effects of fibers on the absorption capacity of RAC [53]. However, the remains a scarcity of information in the literature on the mechanical and permeability characteristics of GFRRAC with different types of mineral admixtures (e.g., RHA, MS, and BFS). Therefore, the main objective of this research work is to examine the effects of potential waste mineral admixtures on the properties of GFRRAC. The studied properties include workability, compressive strength (f_CS_), modulus of elasticity (MOE), splitting tensile strength (f_SP_), modulus of rupture (MOR), water absorption (WA), and rapid chloride ion permeability (RCP). In addition, the synergistic effects of mineral admixtures and glass fibers are also systematically presented for all of the studied properties of RAC. 

The outcome of this research can be used for the evaluation of the effect of local waste mineral admixtures found in Pakistan on the properties of plain and fiber-reinforced concrete. A variety of natural and waste mineral admixtures are cheaply available in Pakistan, e.g., bentonite, metakaolin, rice husk ash, slag, fly ash, and silica fume [13,55,56,57], but there is a lack of published information on the properties of these mineral admixtures in normal and fiber-reinforced concrete. To cope with the rising climate changes in Pakistan and surrounding regions, it is necessary to reduce the reliance of the concrete industry on cement production by supplementing the demand for cement with supplementary mineral admixtures. The findings of the present research will also be useful for the field of sustainable fiber-reinforced concrete development with the use of RCA and waste admixtures.

## 3. Experimental Program

### 3.1. Details of Materials

#### 3.1.1. Binders

The five types of binders used in this cement were Portland cement type I ASTM; uncondensed MS with up to 98.4% silica content; siliceous FA known as type F coal ash; BFS produced in the blast furnace with iron smelting; and RHA produced by calcining rice hull at about 760 °C for 2 h. MS and BFS were obtained from a steel mill that manufactures steel and ferro-silicon alloys. Class-f-type FA was acquired from the Qadirabad coal power plant in Sahiwal, Pakistan. RHA was obtained from a local rice mill in Sahiwal. Unlike BFS, FA, and MS, RHA can be obtained from many rice mills plants in many cities of Pakistan, especially in the Punjab region. RHA has the advantage of lower transportation costs; therefore, it can be expected to produce the smallest carbon footprint among all of the mineral admixtures. BFS, FA, and MS can only be assessed in a couple of major cities (i.e., Karachi and Lahore) of Pakistan. For other cities, BFS, FA, and MS will have high transportation costs and carbon footprints. BFS and RHA have the drawbacks of additional requirements of grinding, unlike FA and MS. Mechanical grinding is necessary to enhance their reactivity in the binder matrix, and, therefore, this can add to the final cost and carbon footprint of RHA and BFS. Both BFS and RHA ground to 45 microns are used as cement replacement. The characteristic properties of all binders are given in Table 1. All chemical properties of binders were measured by X-ray fluorescence (XRF) analysis. The physical appearances of these mineral admixtures are shown in Figure 1.

#### 3.1.2. Aggregates

Siliceous sand of the Lawrancepur quarry was used as the fine aggregate. To prepare a control concrete (CON), natural coarse aggregate (NCA) of crushed limestone was used. RCA was obtained from Margalla-Hills, Taxila, Pakistan. To manufacture RAC, primarily hydrated RCA was obtained by crushing 2-month-old specimens of concrete. The aging of RCA is important for the hydration of adhered mortar and to minimize the chances of latent hydration when used to produce RAC. The compressive strength of these normal-strength concrete samples was about 30–35 MPa. The maximum size of NCA and RCA was selected as 12.5 mm. An overview of RCA is shown in Figure 2. The properties of the aggregates are shown in Table 2. A granulometry analysis report of the aggregates is illustrated in Figure 3. Gradation (variation in particle sizes) of both natural and recycled aggregates is kept the same. This was achieved by mixing the retained masses of coarse aggregates using a 12.5, 9.5, 4.75, and 2.36 mm sieve according to gradation shown in Figure 3.

#### 3.1.3. Glass Fibers

Microglass fibers used in this research are high tensile strength alkali-resistant chopped strands. These fibers had filaments of lengths varying between 12 and 18 mm and a diameter of 15 µm. The tensile strength of glass fibers was 1500 MPa. The density of these fibers was 2600 kg/m^3^. An overview of glass fiber is shown in Figure 4.

#### 3.1.4. Superplasticizer and Water

“Viscocrete-3110”, (Rawalpindi, Punjab, Pakistan) a third-generation water-reducer, was used as a superplasticizer (SP) to control slump loss due to MS and fiber addition. Tap water was utilized in both the preparation and curing of concrete specimens.

### 3.2. Composition of All Concrete Mixtures

In this research, one control mix (CON) and two types of RAC families were produced with and without glass fibers. The CON concrete was devised for compressive strength (f_CS_) of 40 MPa following the ACI mix design guidelines [58]. The selected water–cement ratio was 0.38. In CON mix, NCA was replaced with RCA to produce plain RAC (PRAC). To study the effects of waste mineral admixtures on properties of PRAC, MS, BFS, RHA, and FA were used as cement replacements by volume. The replacement levels of MS, FA, BFS, and RHA were 10%, 15%, 30%, and 15%, respectively by volume of cement. These levels of mineral admixtures were selected based on the optimum results of compression testing at 28 days in recent research [59,60,61]. We used a 1% glass fiber as a volume fraction of RAC. The nomenclature of all mixes is given in Table 3.

The composition details of the studied mixes are shown in Table 4. To satisfy the high water demand of RCA, it was presoaked in tap water for a duration of 1 h. This is necessary, because, otherwise, air-dried or oven-dried RCA absorbs the water from the binder matrix and decreases the effective water–cement ratio and workability. Prior to mixing, pre-soaked RCA was air dried for 15 min to remove the surface moisture. The workability of all mixes was required in the range of 70–100 mm. Mixes involving MS and fibers required a superplasticizer to compensate for the fluidity loss. Although BFS, RHA, and FA do not affect the workability of fresh mixes, BFS and FA were useful in enhancing the workability of PRAC and GGFRAC. The doses of fibers and plasticizers are given in Table 4.

All concrete mixes were produced in a rapid speed machine mixer. In the first stage, aggregates and binders were dry melded at 60 rev/min for 2 min. Subsequently, ½ quantity of water and superplasticizer was added to the mix, and blending continued at the speed of 80 rev/min for 2 min. In the last stage, fibers and the remaining ½ water and plasticizer were added to mix and melded at the constant speed of 100 rev/min for 4 min.

### 3.3. Preparation of Specimens for Testing

All samples were molded in standard-sized steel casts. After casting, the fresh samples were allowed to set for 24 h under polythene sheets to avoid moisture loss from their surface. The samples were removed from the casts after 24 h and subsequently cured in tap water for 28 days. After curing, the specimens were conditioned in air for 2 days and finally subjected to mechanical and durability testing.

#### 3.3.1. Compression Testing

To evaluate the influence of mineral admixtures on the compressive strength of PRAC and GFRRAC mixes, 100 mm × 200 mm cylindrical samples were tested as per ASTM C39 [62]. The samples were tested in a CONTROLS-UTM machine with a compression loading capacity of 3000 kN at a displacement rate of 0.005 mm/sec.

#### 3.3.2. Modulus of Elasticity Testing

The modulus of elasticity was measured using Equation (3) as given by the standard ASTM C469. The compression load deformation characteristics of the samples (i.e., 100 mm × 200 mm concrete cylinders) were obtained using a data acquisition system (DAS), where loads were registered against axial deflection. The axial deflection was measured using LVDTs. The LVDTs were installed at four points on the specimen at an angle of 90° to each other. The clear distance between LVDTs was 100 mm. All samples were deformed at a displacement rate of 0.005 mm/sec. Recorded compression loads v/s axial deflections were used to calculate the modulus of elasticity of the specimens using Equation (1).
(1)$$MOE=\frac{f_{2}-f_{1}}{\in_{2}-\in_{1}}$$
where MOE is modulus of elasticity (MPa), ∈_1_ is the compressive strain of 0.00005, and f_1_ is compressive stress (MPa) related to ∈_1_. In addition, f_2_ is compressive stress equal to 0.4 × peak stress and ∈_2_ is the strain value related to f_2_.

#### 3.3.3. Splitting Tensile Testing

To calculate the indirect tensile strength of mixes, 100 mm × 200 mm samples were tested as per ASTM C496 [63]. This test was performed in the CONTROLS-UTM. The tensile strength of samples was measured by using Equation (2).
(2)$$f_{sp}=\frac{2000\times P}{\pi \times D\times L}$$
where f_SP_ is splitting tensile strength (MPa); P is the maximum load at sample (kN); and D and L are the diameter (mm) and length (mm) of the sample, respectively.

#### 3.3.4. Bending Testing 

To study the modulus of rupture capacity of PRAC and GFFRAC, prismatic specimens with dimensions of 100 mm × 100 mm × 500 mm were subjected to a third-point bending test according to ASTM C1609 [64]. Prismatic samples were stressed under a loading rate of 1 MPa/min. The modulus of rupture was determined using Equation (3).
(3)$$MOR=\frac{3\times P\times L}{2\times B\times D^{2}}$$
where MOR is the modulus of rupture (MPa); P is the maximum load at failure of the prism (N)l and L, B, and D, are the length, breadth, and height of prismatic sample (mm), respectively.

#### 3.3.5. Water Absorption Capacity Testing

To investigate the influence of mineral admixtures on the absorption capacity of PRAC and GFRRAC, 100 mm × 100 mm cubical specimens of each mix were prepared. These specimens were tested according to ASTM C948 [65] to obtain the water absorption capacities of plain and glass fiber-reinforced composites.

#### 3.3.6. Chloride Penetration Testing

The chloride permeability resistance of PRAC and GFFRAC mixes was tested following ASTM 1202. This test was performed on specimens with a thickness of 50 mm and diameter of 100 mm cut from the concrete cylinder with dimensions of 100 mm × 200 mm. Chloride ion penetration is represented by the electrical charge passed through 50 mm × 100 mm discs during a 6 h period.

## 4. Results of Testing and Discussion

### 4.1. Workability

The workability of fresh concrete is a major issue that can occur during the application of RCA and fibers in concrete preparation. The results of slump testing of all mixes with and without a superplasticizer dose are shown in Figure 5. RCA did not show a detrimental effect on workability. This is because the water absorption capacity of RCA was satisfied by presoaking in tap water prior to the preparation of mixes. Therefore, RAC shows workability well within the range of the target slump of 70–100 mm. 

Fresh RAC mixes containing glass fibers show a slump value of zero. This is because of the balling effect of fibers in fresh concrete. Fibers, when added to fresh concrete, convert the fresh mix into lumps or small balls. Small lumps are formed due to the increased integrity of fresh concrete. Therefore, to achieve the desired workability, superplasticizer (SP) is added to GFRRACs. As shown in Figure 5, the slump of 86 mm was achieved when SP was used at the dosage of 0.7% by mass of the binder.

The effect of mineral admixtures considerably varies between the types of admixtures. MS-added RAC showed zero slump without SP. This is because of the very high specific surface area of MS particles. Moreover, MS has been observed to increase the cohesion of fresh concrete, which consequently adversely affects workability [66]. To compensate for slump loss due to MS, 0.5% SP was added to RAC by mass of cement. Partial replacement of cement with BFS, RHA, and FA did not affect the workability of fresh concrete. This is because these mineral admixtures have almost the same fineness as that of cement. FA and BFS showed a slightly higher slump than PRAC, which is probably because of the slow reactivity of particles of these binders in the concrete matrix. The slower reactivity of BFS and FA particles saves the free water in the binder matrix, which is helpful in the lubrication and good packing of concrete particles. FA is useful in lowering the water demand of concrete [67]. Similarly, BFS is also a good water reducer [68]; thus, RAC with 30% replacement of cement with BFS shows a higher slump than RAC without slag.

The adverse workability without SP was observed in GFRRAC-10MS since both MS and glass fibers are detrimental to the flow of concrete. Therefore, a high SP dosage of 1% was used in GFRRAC-10MS to achieve a slump in the range of 70–100 mm. Unlike MS, both BFS and FA reduced the SP dosage required for GFRRAC to reach the target slump. This was due to the positive effect of BFS and FA on workability. The slump testing results show the usefulness of BFS and FA in reducing the cost of plasticizers in GFRRAC. RHA showed an insignificant role in workability. In fact, RHA and OPC-based mixes showed similar workability.

### 4.2. Compressive Strength

The compressive strength (f_CS_) of each mix at the age of 28 days is shown in Figure 6. As expected, PRAC shows lower f_CS_ than that of CON mix. The f_CS_ of RAC is 17% lesser than that of CON mix. This is predominately because of the high porosity and water content of RAC. RCA contains pores due to the attached mortar, and it absorbs a large amount of water in the presoaking period; therefore, for an equivalent water-to-binder ratio, RAC has a higher net volume of water and lower f_CS_ compared to CON mix. 

The 1% glass fiber addition to RAC leads to an insignificant increase in f_CS_. GFRRAC shows 6.2% more f_CS_ than that of PRAC. The addition of glass fibers alone to RAC does not help in recovering f_CS_ loss due to the full replacement of NCA with RCA. Minor improvements in f_CS_ due to fibers are generally attributed to the increase in the stiffness of the concrete matrix due to the confinement effect of glass fibers. Fiber reinforcement also delays the onset of peak load and controls the proliferation of micro- and macro-cracking of the concrete matrix. Previous studies [48,49,52,69] on glass, steel, basalt, and polypropylene fibers have also observed that fiber addition is not very helpful in upgrading the f_CS_ of concrete. Kizilkanat et al. [48] reported an f_CS_ increase of 4.7% in high-strength concrete due to 1% glass fiber addition. 

The net change in the f_CS_ of PRAC and GFRRAC due to the addition of mineral admixtures is shown in Figure 7. Generally, all mineral admixtures increase the 28-day f_CS_ of PRAC and GFRRAC differently. Mineral admixtures are attributed to f_CS_ by three different actions: (1) the pozzolanic reaction that converts free portlandite into strong calcium-silicate hydrate (CSH) gel; (2) the filling effect of mineral particles reducing the volume of voids in concrete by filling spaces between cement particles that results in an improvement in concrete density and strength; (3) and some mineral admixtures (i.e., BFS and FA) attributed to f_CS_ by reducing the water demand, thereby increasing the available quantity of water in the binder matrix for the hydration of cement particles, thus indicating that these binders can help in reducing microcracks due to the release of hydration heat in cement-rich concretes (i.e., high-strength concrete).

MS is superior to BFS, FA, and RHA in upgrading the f_CS_ of RAC. Followed by MS, RHA also shows promising results in enhancing the f_CS_ of RAC. Unlike glass fiber, both MS and RHA can significantly advance the f_CS_ of RAC alone. MS is highly effective in the strengthening of the binder matrix because its ultra-fine reactive particles consume portlandite in pozzolanic reactions faster than both FA and BFS. RHA is also rich in active silica, which helps in faster pozzolanic reactions [40]. The f_CS_ increment due to the filling action of mineral particles is dominant in the MS-containing mixtures. In addition to the filling action and pozzolanic reactions, BFS contains high amounts of lime, silica, and alumina, which also form the strength developing hydration products similar to cement particles but at a slower pace. FA shows lower f_CS_ than that of the other admixtures, mainly because of its high silica and alumina contents and low calcium oxide content that reduces the pozzolanicity of concrete. FA may also contain non-reactive silica that requires a longer duration for the development of full strength [70].

Although individual mineral admixtures and glass fibers fail to overcome the loss of f_CS_ due to the incorporation of RCA into concrete, their combined incorporation can significantly help in upgrading the f_CS_ of RAC compared to CON mix. Figure 8 shows the effect of the combined incorporation of mineral admixtures and glass fibers on the net change in f_CS_ of RAC. GFFRAC-MS shows an f_CS_ increase of about 3.5% when compared with CON mix, and the f_CS_ of GFFRAC-RHA becomes sufficiently close to that of CON mix.

Figure 8 also shows that combining glass fibers and mineral admixtures does not only combine their individual benefits, but it also shows an extra net increase in f_CS_ due to their synergistic behavior. Mineral admixtures can improve the utilization of fibers by strengthening the binder matrix. Improvement in the bond strength between the glass fibers and binder matrix increases the efficiency of fiber filaments in resisting compressive loadings. The maximum synergistic effect is observed when glass fibers are used with MS. This is because MS is an excellent supplementary cementitious material compared to BFS, FA, and RHA. Studies have shown that stress transfer between the binder matrix and fibers increases when mineral admixtures are used as cement replacement [51,52,71,72]. Mineral admixtures also improve the dispersion of reinforcement material to avoid clustering of fibers [73,74], thereby increasing the efficiency of fibers.

### 4.3. Modulus of Elasticity

The modulus of elasticity (MOE) of PRAC and GFRRAC mixes is shown in Figure 9. The MOE of concrete drops by about 13% when RCA is used instead of NCA in concrete. As RCA is inferior in strength compared to NCA, due to the presence of old mortar, RAC shows a lower MOE than that of CON mix.

The 1% glass fiber incorporation increases the MOE by 4.2%—see Figure 9b. Fibers increase the compressive stiffness of concrete due to their confinement effect. Moreover, fibers have a very high elastic modulus than that of plain concrete; therefore, the composite effect of glass fibers and concrete can result in an overall improvement in the MOE of product concrete. Kizilkanat et al. [48] reported an insignificant effect of glass fibers on the MOE of concrete, and similar results were reported by Tassew and Lubell [75]. Thomas and Ramaswamy [76] showed that the addition of steel fiber improves the MOE by less than 10%. Fibers are known for their mixed effects on the MOE and f_CS_. Ali et al. [77] showed that fibers have both positive and negative effects on compressive properties at different doses. They showed that at doses lower than 1%, fibers improved the f_CS_ and MOE, whereas at higher doses, fibers were detrimental to the compressive stiffness of concrete due to introduction of a large number of interfacial transition zones. Figure 9 shows that mineral admixtures prove to be more useful than glass fibers in advancing the MOE of RAC. The highest improvement occurs with the addition of MS. PRAC-10MS shows an MOE that is only 5% lower than that of the conventional CON mix. Similar to f_CS_, filling effects and pozzolanic reactions also contribute to the MOE. 

The combined incorporation of glass fibers and mineral admixtures also yields synergistic effects on the MOE. For example, individually, MS and glass fibers show an improvement of 8.7 and 4.2%, respectively, but their combined use yields a net gain of 16.5%. The coupling effect of MS and fibers yields 3.6% more net gain in the MOE than the anticipated net gain of 12.9%. Koksal et al. [78] have confirmed the coupling effects of MS and fibers on compressive properties. Ali and Qureshi showed that glass fiber shows an improvement in bond strength with FA. Since both the MOE and f_CS_ are similarly affected by the incorporation of mineral admixtures and fiber, both of these parameters are correlated with high accuracy, as shown in Figure 10. This shows that the effect of the addition of mineral admixtures and fibers on the f_CS_ can be used to predict the change in the MOE. 

### 4.4. Splitting Tensile Strength

The effect of mineral admixtures (BFS, SF, FA, and RHA) and glass fibers on the 28-day splitting tensile strength (f_SP_) of RAC mixes is shown in Figure 11. As expected, RAC showed 13.5% less f_SP_ than that of CON mix. This is already ascribed to the inherited weakness of RAC due to its high porosity. Mineral admixtures help in compensating some of f_SP_ loss due to the full replacement of NCA with RCA. In contrast, RAC, with the help of glass fiber, beats the conventional CON mix by a huge margin.

RAC experiences f_SP_ improvement of 29% due to 1% glass fiber addition. This shows that incorporating glass fibers alone is sufficient enough for the f_SP_ of RAC to surpass that of CON mix. Fiber addition is extremely helpful in upgrading the tensile strength of concrete. When concrete is pulled apart under the splitting action of loading, fibers activate earlier than the peak load; therefore, they delay the onset of matrix failure, which increases tensile strength. Due to their high tensile strength, fibers can overcome the issue of brittleness of a plain concrete matrix. In normal-strength concrete, Ali and Qureshi [12] reported an increase of 22% in f_SP_ at 1% glass fiber. This improvement is lower than that observed in the present study. This might be because of the higher binder matrix strength in the present study. Abbass et al. [79] showed that fibers show more efficiency in high strength classes. The results of both compression and splitting tensile testing showed that fiber addition is more helpful in tensile strength. Therefore, fibers should be used when the goal is to obtain high ductility, as they show uneconomical gains in compressive strength [50,80].

Figure 12 shows the effect of mineral admixtures on the net change in f_SP_ of PRAC and GFRRAC. Mineral admixtures show 1.5–11% improvement in f_SP_ of RAC depending on their type. The highest improvement of 11% is observed with MS, which is followed by RHA with an improvement of over 6%. PRAC-10MS almost reaches the potential of CON mix. These results show that both RHA and MS can be used to enhance the f_SP_ of RAC. In contrast, BFS and FA are not useful in increasing the tensile strength of RAC. Higher reactivity of MS and RHA compared to that of BFS and FA is explained in the discussion of the results of compression testing. The behavior of f_SP_ and f_CS_ with the variation in mineral admixtures is almost identical. This is because the development of a microstructure due to pozzolanic reactions and filling effects contributes similarly to both f_SP_ and f_CS_. Koksal et al. [78] reported that silica fume similarly enhances both f_SP_ and f_CS_. This behavior is also observed to be true for other mineral admixtures, i.e., FA [53], RHA [40], and BFS [81].

Figure 13 shows the combined effect of glass fiber and mineral admixture incorporation on the f_SP_ of RAC. In the combined effect, the role of glass fibers is dominant over that of the mineral admixtures. This analysis shows that with the combination of both glass fibers and mineral admixtures, an overall improvement of 34–50% is obtained for the f_SP_ of RAC. The 1% glass fiber + 10% MS shows superior performance over that of all the other combinations. GFRRAC-10MS shows 30% more f_SP_ than that of CON mix. The synergistic effects due to the combined incorporation of MS and glass fibers are also better than those of the other combinations. An extra net gain of about 10% is achieved when MS and glass fiber are used together. The combinations of FA and BFS with glass fibers show a lower synergistic effect compared to that of the combination of MS and RHA. This is because MS- and RHA-added binders have higher strength compared to that of those with FA and BFS. The synergistic effect is solely due to the improvement in the bond strength of fibers with a concrete matrix. In a weaker matrix, fibers are easily slipped and pulled under tensile loading without the utilization of their high tensile strength, whereas in a stronger binder matrix, fibers are highly stretched due to a strong grip of the binder matrix, which consequently improves the tensile capacity of concrete. It can be stated that mineral admixtures advance the efficiency of fiber reinforcement.

### 4.5. Modulus of Rupture

The modulus of rupture (MOR), also known as flexural strength, is an estimate of the tensile strength of concrete. The MOR is used instead of compressive strength in the thickness design of pavements and thin shell elements. The effect of mineral admixtures on the 28-day MOR of PRAC and GFRRAC is shown in Figure 14. The MOR is reduced by 10% due to the replacement of NCA with RCA. Loss of the MOR due to RCA incorporation is lower compared to that observed in f_CS_. This is because the angularity of RCA is higher than that of NCA; therefore, its angularity compensates some loss in the MOR. This finding is in line with that of Ahmadi et al. [28].

The effect of the addition of mineral admixtures and glass fibers on the net change in the MOR of RAC is shown in Figure 15. Glass fibers proved to be more useful in advancing the MOR than it was in the f_CS_ and MOE. With the addition of glass fiber, RAC shows a 41% increase over plain RAC. GFRRAC shows an MOR that is 29% higher than that of conventional CON mix. These results show that in order to overcome the issue of brittleness and low tensile strength of RAC, fiber reinforcement is a good solution. Research on other fibers, e.g., steel, polypropylene, and carbon fibers [71,82], has also established that fiber addition is more useful to the MOR compared than f_CS_ and f_SP_. 

The effect of mineral admixtures on the MOR is very similar to that observed in other mechanical properties, i.e., the f_CS_, MOE, and f_SP_. MS addition provides the best results among all of the mineral admixtures. MS alone is sufficient to overcome the MOR loss of RAC w.r.t CON mix. Moreover, other admixtures are also useful in closing the gap between the MOR values of RAC and CON mix. Hardly any difference is observed in the performance of RAC with BFS, RHA, and FA. These improvements are ascribed to the microstructural developments due to pozzolanic reactions and pore refinement in the binder matrix.

The combined effect of the glass fibers and mineral admixtures on the MOR of RAC is shown in Figure 16. In the combined effect, the contribution of fiber reinforcement is dominant. Since fibers add about 40% to the MOR of RAC, a large proportion of improvement is contributed by the fibers. The efficiency of fiber reinforcement improves with all mineral admixtures. GFRRAC shows a 20% increase in the MOR with MS compared to GFRRAC without MS. GFRRAC also shows a noticeable synergistic effect due to the addition of RHA. With RHA, 15% improvement in net gain is observed due to the addition of glass fibers in RAC. Both BFS and FA show minimal synergistic effects probably due to their lower binder strength compared to that of RHA and MS.

The correlation between the MOR and f_CS_ and f_SP_ is shown in Figure 17. The measurement of the MOR requires careful preparation and testing of the prismatic specimen; therefore, to avoid any field discrepancies, the MOR is usually assessed from the f_CS_. For plain concretes, relationships and technical standards that can fairly predict the MOR from f_CS_ have been developed in the literature. However, for fiber-reinforced concretes, it is still difficult to assess the MOR from f_CS_ without considering the effect of fibers, e.g., fiber dose, and type. [53,57]. This is because fibers do not significantly contribute to f_CS_, but they show beneficial effects on the MOR and other tensile strength parameters. Thus, Figure 17 shows that f_CS_ is weakly related to the MOR. Therefore, correlations are derived separately between the f_CS_ and MOR of plain and fiber-reinforced concretes. On the other hand, it is simpler to precisely evaluate f_SP_ than the MOR, as f_SP_ requires higher loading unlike the MOR test to fail the specimen under splitting action of loads. Moreover, the effect of fibers is almost identical on both the f_SP_ and MOR. Therefore, the MOR can be predicted accurately from the f_SP_, as shown in Figure 17b. These correlations show that to estimate the MOR without considering the effect, f_SP_ test results can be used instead of f_CS_.

### 4.6. Water Absorption

The effect of mineral admixtures and glass fibers on the 28-day water absorption (WA) capacity of RAC is shown in Figure 18. WA is the measure of the penetrable volume of concrete. Most of the harmful chemicals (slats, alkalis, carbon dioxide, acids, etc.) penetrate into the concrete through permeable voids; therefore, WA is an indirect and simple assessment of concrete’s durability. RAC concrete shows 30% higher WA compared to that of NAC, i.e., CON mix. Pores present in the old mortar of RCA increase the capillarity capacity of the concrete matrix. In contrast, the addition of glass fibers causes a slight WA increase of 2.3% in RAC, as shown in Figure 19. This shows that despite the useful results for mechanical performance, glass fibers can slightly increase the porosity of concrete. Similar findings have been observed for polypropylene and glass fibers in previous studies [47,53]. Microfibers are suspected to increase the connectivity of permeable porosity of the concrete matrix. However, this behavior is different compared to that of macrofibers such as steel fiber, which shows a slight improvement in the permeability resistance of concrete at a 1% volume fraction [52].

The addition of mineral admixtures significantly reduces the WA of RAC, as shown in Figure 19. PRAC-10MS shows lower WA than that of CON mix. Moreover, all mineral admixtures reduce the WA capacity by 12–24%. The addition of BFS and FA proves to be more useful in reducing WA than their effect on mechanical properties. This is ascribed to their filler effect and capability to reduce the heat of hydration. The filler effect reduces the pore size, while a reduction in the heat of hydration can control shrinkage and consequent microcracks in the binder matrix. 

Both glass fibers and mineral admixtures show the opposite effects on WA when used individually in RAC. In regard to combined effects, it can be observed that both glass fibers and mineral admixtures have positive effects on reducing the WA capacity of RAC. For example, combined incorporation of glass fibers and MS was expected to yield a net reduction of 21.3%, but experimentally, a WA reduction of 26.3% was obtained with 1% glass fiber + 10% MS addition, which is almost 5% higher than the expected reduction in WA. GFRRAC with MS shows a lower WA than that of CON. Moreover, minor synergistic effects can be observed on the WA reduction due to the combined use of other mineral admixtures (i.e., BFS, FA, and RHA) and fiber—see Figure 20. This synergistic effect can be ascribed to the improvement in the interfacial transition zone between the fibers and binder matrix. Moreover, fibers can minimize the negative effects of the shrinkage cracking on the WA capacity of plain concrete [52]. 

### 4.7. Chloride Penetration

Figure 21 shows the 28-day rapid chloride ion penetration (RCP) of PRAC and GFRRAC mixtures. RAC has 8% lower RCP resistance than that of CON mix. The addition of glass fibers causes insignificant changes in the RCP of RAC. However, the RCP resistance of RAC with a mineral admixture is generally improved compared to that of CON.

Unlike WA results, a notable improvement in RCP resistance can be observed with the addition of FA and BFS. The RCP of RAC decreases by more than 16% due to the addition of BFS or FA, whereas RHA and MS show a minimal improvement of 5–9.7% in RCP resistance compared to both BFS and MS. This is because high alumina content leads to the formation of tricalcium aluminates that are more resistant to RCP. Uysal and Yilmaz [83] reported that charge passed through specimen reduces with the rise in alumina content of the binder. This is the reason why slag is used in the manufacturing of sulfate-resisting cements. Kou et al. [18] showed that BFS and FA have a more positive effect than that of MS on the RCP resistance of concrete. Due to pozzolanic admixtures, GFRRAC shows RCP resistance that is better than or comparable to that of CON mix. Minimal permeability can be observed for GFRRAC-FA and GFRRAC-BFS mixes.

RCP and WA of RAC can be correlated with each other (see Figure 22), because both of these parameters depend on the degree of hydration of the binder, microstructural porosity, and pore refinement. An increase in pore size or poorly developed cementing compounds directly increases the RCP and WA capacity of concrete. Therefore, these parameters can be predicted from each other, as shown in Figure 22. WA measurement is relatively simpler than RCP; therefore, RCP can be predicted from WA as a rough estimate of the chloride durability of concrete. Moreover, RCP and WA can also be correlated with f_CS_, as it is sensitive to changes in pore volume. The f_CS_ increases with the reduction in porosity and vice versa. Therefore, change in f_CS_ can be used as a direct indicator of lower or higher permeability resistance. RCP and WA are inversely related to f_CS_.

## 5. Conclusions

The following conclusions can be derived from the experimental results:

The slump test results show that BFS and FA are useful in enhancing the workability of concrete. FA and BFS reduce the superplasticizer demand that is increased due to the addition of glass fibers in RAC. MS has detrimental effects, and the workability of concrete worsens with the combined addition of silica and microglass fibers. Therefore, RAC incorporating both fibers and microsilica requires a high dose of superplasticizer to achieve the desired workability.

FA and BFS show minor improvements in the mechanical performance of RAC. Along with fibers, the combination of FA and BFS upgrades the compressive strength (f_CS_) of RAC by 12–17%, whereas MS and RHA produce the highest net gains in f_CS_ of RAC with and without fibers. GFRRAC-10MS and GFRRAC-15RHA show a net improvement of 23.5 and 19% in f_CS_ when compared with that of PRAC, respectively. The f_CS_ of GFRRAC-10MS exceeds that of the conventional aggregate CON mix. 

The modulus of elasticity (MOE) is affected similarly to f_CS_ due to the individual and combined incorporation of mineral admixtures and fibers. GFRRAC-10MS shows an MOE that is 16 and 3% higher than that of CON mix and PRAC, respectively.

Generally, the addition of fibers is more useful than that of mineral admixtures in advancing the splitting tensile strength (f_SP_) and modulus of rupture (MOR) of RAC, whereas mineral admixtures yield more net improvements in the f_CS_ and MOE than fibers.

The best synergistic effects of fibers and mineral admixtures can be observed in the results of the f_SP_ and MOR. MS and RHA increase the efficiency of glass fibers by 20 and 15%, respectively, in advancing the f_SP_ of RAC. The combined incorporation of MS and fibers improves the f_SP_ of RAC by more than 50%. GFRRAC-15RHA shows 40% more f_SP_ than that of PRAC.

The MOR highly benefits from the incorporation of fibers compared to the other tested mechanical properties. GFRRAC-10MS and GFRRAC-15RHA show an MOR that is 60 and 55% higher than that of PRAC, respectively. Using fibers is more useful in achieving a higher f_SP_ or MOR than using mineral admixtures for the same purpose.

Mineral admixtures, especially MS and RHA show prominent reductions of 23 and 18%, respectively, in water absorption (WA), while fibers yield the opposite effect and slightly increase the WA of RAC. The addition of BFS and FA to RAC shows a WA reduction of 13–15% w.r.t PRAC. Due to synergistic effects, the combination of fibers and mineral admixtures reduces the WA more than the sum of their single effects. GFRRAC-10MS shows a WA that is 26% lower than that of PRAC. 

BFS and FA are more useful than MS and RHA in declining the rapid chloride penetration (RCP) in RAC. The addition of fibers shows a slightly negative effect on the RCP resistance of RAC. However, the combined incorporation of any of the mineral admixtures with glass fibers can significantly lower the RCP of RAC compared to PRAC. With the addition of BFS, FA, or MS, RAC shows lower RCP than that of conventional CON mix.

## Figures and Tables

**Figure 1 materials-14-05933-f001:**
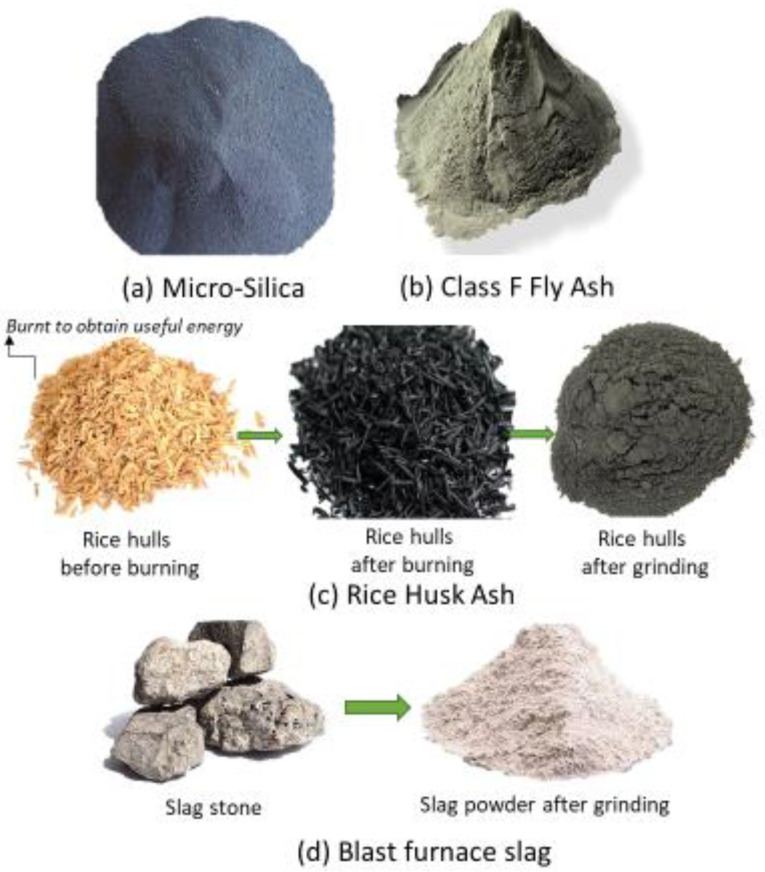
Physical appearances of (**a**) MS, (**b**) FA, (**c**) RHA, and (**d**) BFS.

**Figure 2 materials-14-05933-f002:**
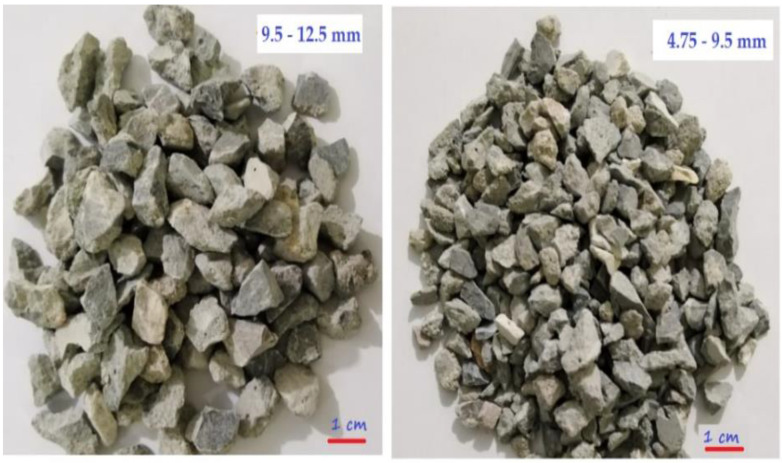
Physical overview of RCA.

**Figure 3 materials-14-05933-f003:**
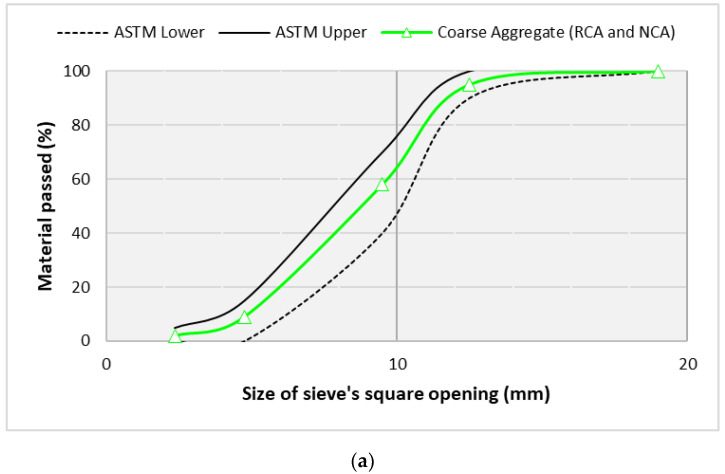
Granulometry of (**a**) coarse aggregates (i.e., NCA and RCA) and (**b**) fine aggregates.

**Figure 4 materials-14-05933-f004:**
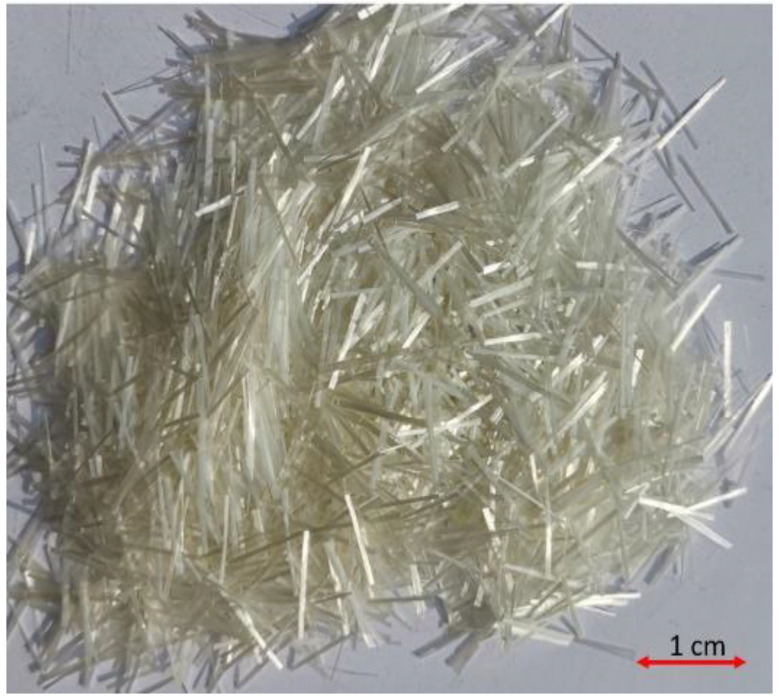
Glass fibers.

**Figure 5 materials-14-05933-f005:**
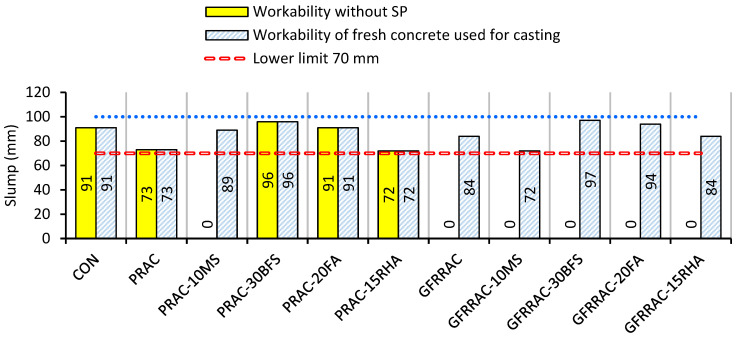
Workability test results of RAC with glass fibers and mineral admixtures.

**Figure 6 materials-14-05933-f006:**
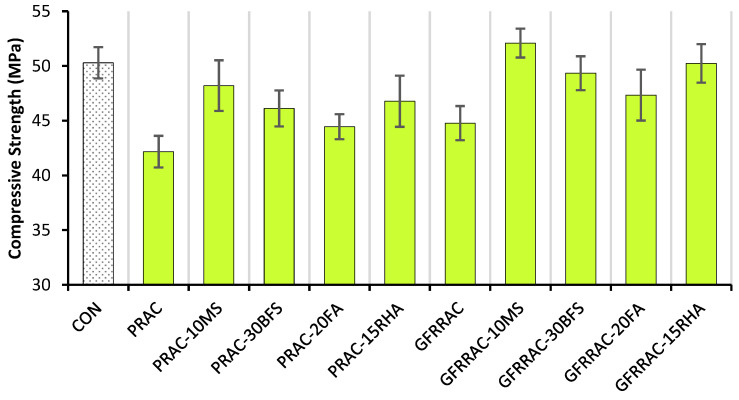
Compressive strength (f_CS_) of PRAC and GFRRAC with different mineral admixtures.

**Figure 7 materials-14-05933-f007:**
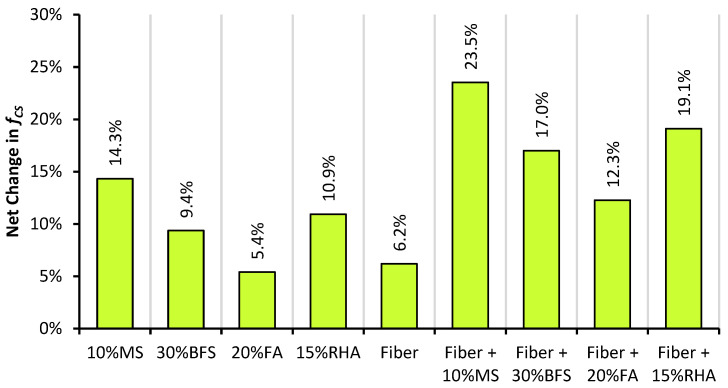
The net change in f_CS_ of PRAC and GFRRAC due to the addition of mineral admixtures (MS, BFS, FA, and RHA).

**Figure 8 materials-14-05933-f008:**
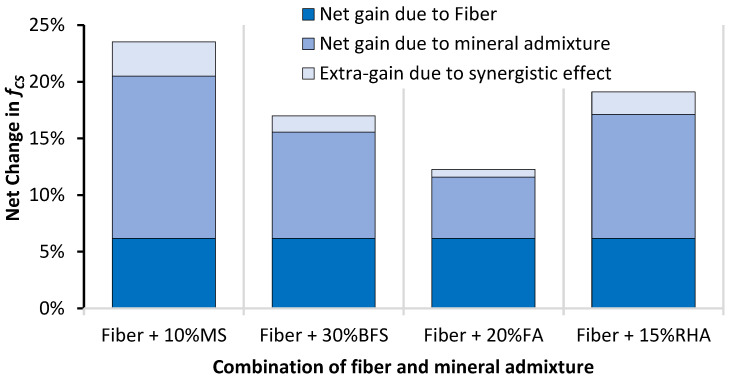
Effect of combined incorporation of mineral admixtures and glass fibers on the net change in f_CS_ of RAC.

**Figure 9 materials-14-05933-f009:**
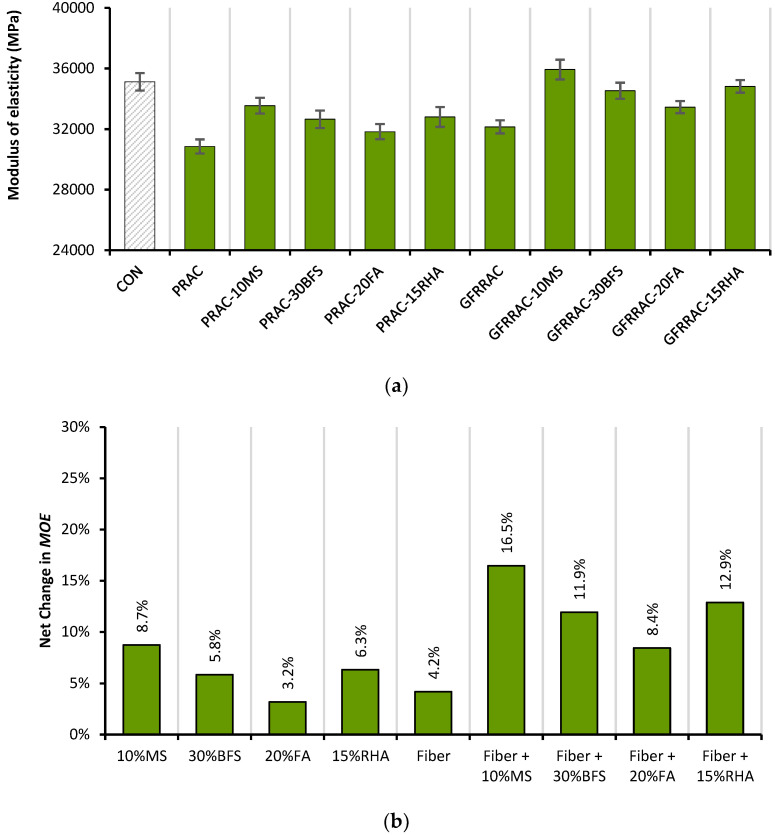
Modulus of elasticity (MOE): (**a**) MOE of PRAC and GFRRAC mixes with varying mineral admixture; (**b**) net change in MOE of RAC due to incorporation of mineral admixtures and glass fibers.

**Figure 10 materials-14-05933-f010:**
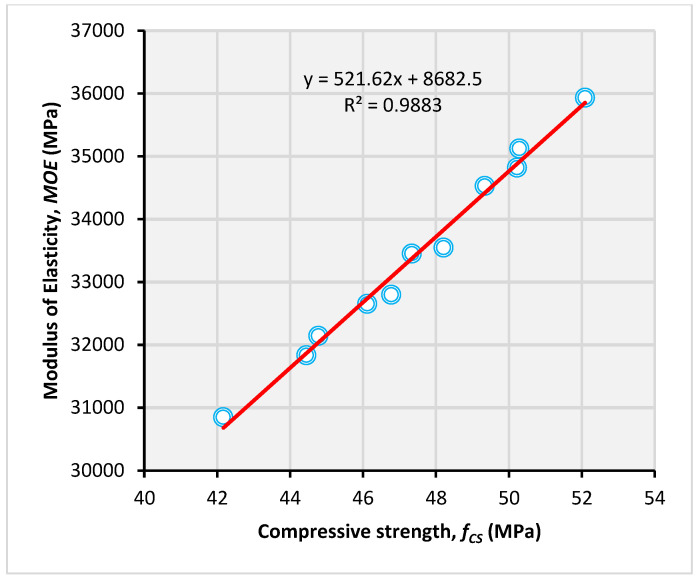
Correlation between f_CS_ and MOE.

**Figure 11 materials-14-05933-f011:**
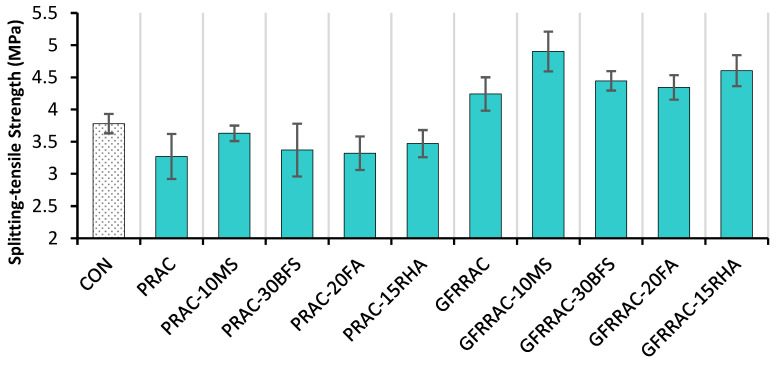
Effect of mineral admixtures and glass fibers on splitting tensile strength (f_SP_) of RAC.

**Figure 12 materials-14-05933-f012:**
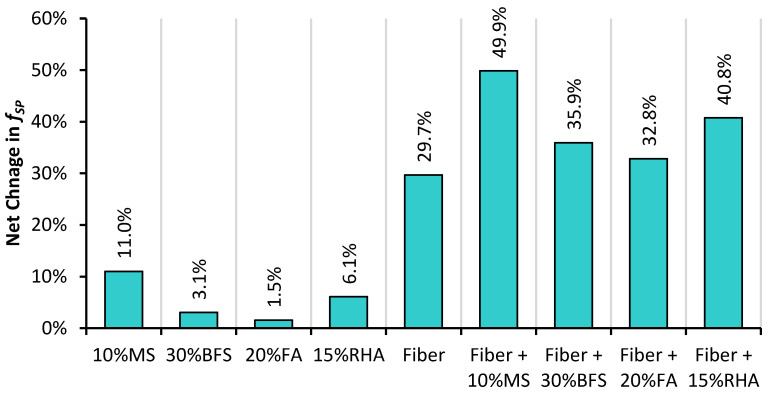
The net change in f_SP_ of PRAC and GFRRAC due to the addition of mineral admixtures (MS, BFS, FA, and RHA).

**Figure 13 materials-14-05933-f013:**
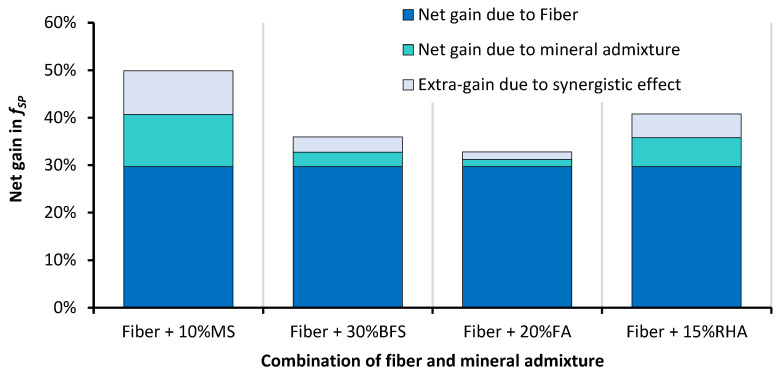
The effect of the combined incorporation of mineral admixtures and glass fibers on the net change in f_SP_ of RAC.

**Figure 14 materials-14-05933-f014:**
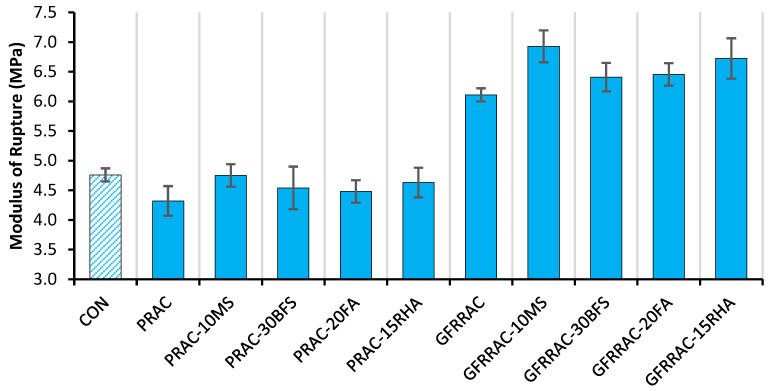
The effect of mineral admixtures and glass fibers on the modulus of rupture (MOR) of RAC.

**Figure 15 materials-14-05933-f015:**
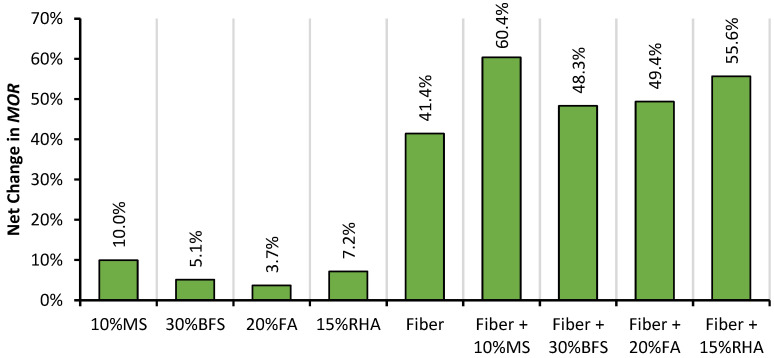
The net change in the MOR of RAC due to the addition of mineral admixtures and glass fibers (MS, BFS, FA, and RHA).

**Figure 16 materials-14-05933-f016:**
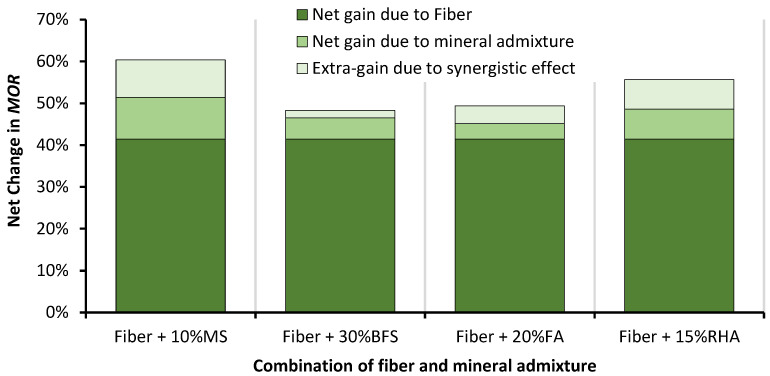
The net change in the MOR of RAC due to the combined effect of mineral admixtures and fibers.

**Figure 17 materials-14-05933-f017:**
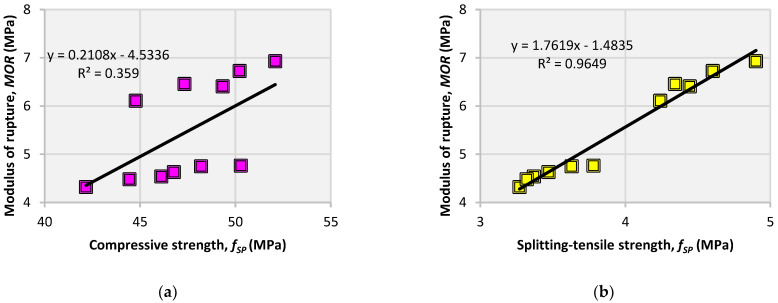
Correlation between (**a**) MOR and f_CS_; (**b**) MOR and f_SP_; and (**c**) MOR, f_SP_, and f_CS_.

**Figure 18 materials-14-05933-f018:**
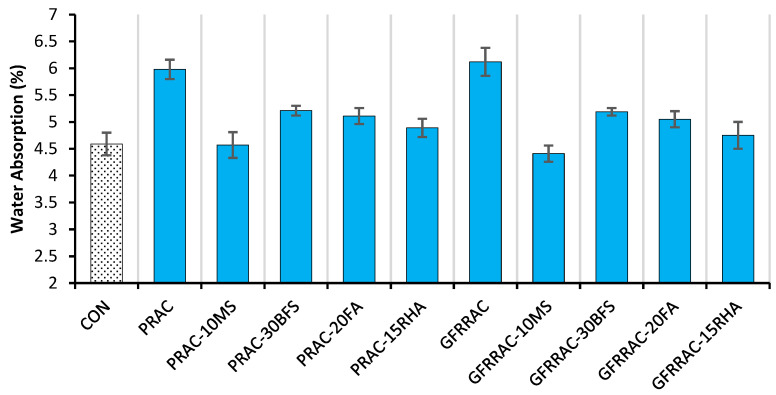
Water absorption capacity of RAC with different mineral admixtures and glass fibers.

**Figure 19 materials-14-05933-f019:**
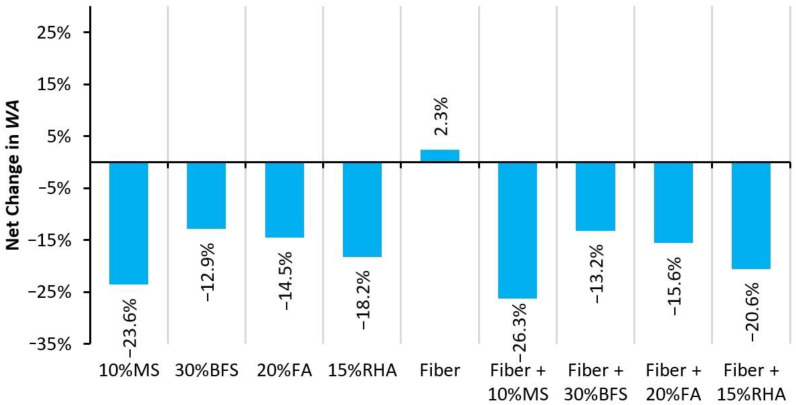
Net change in WA due to the addition of fibers and mineral admixtures.

**Figure 20 materials-14-05933-f020:**
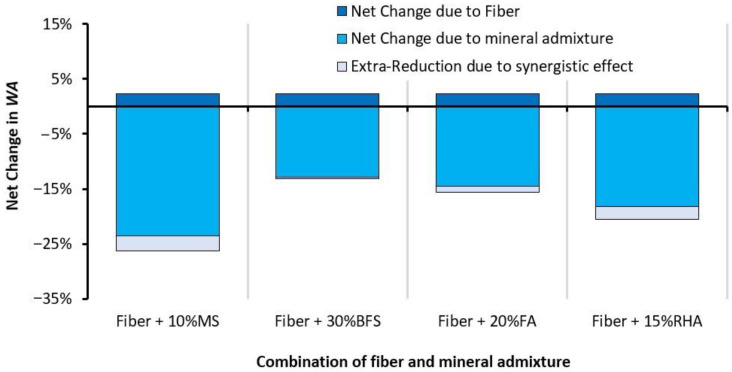
The combined effect of mineral admixtures and fibers on the net change in the WA of RAC.

**Figure 21 materials-14-05933-f021:**
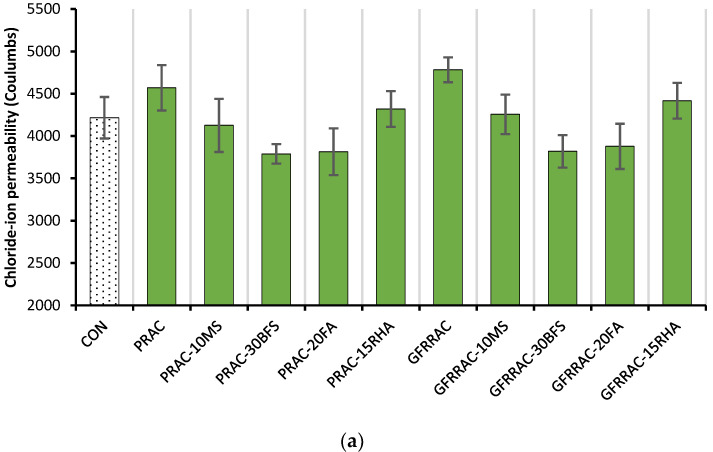
Rapid chloride ion penetration (RCP): (**a**) RCP of PRAC and GFRRAC mixtures with different mineral admixture; (**b**) net change in RCP due to the addition of fibers and mineral admixtures.

**Figure 22 materials-14-05933-f022:**
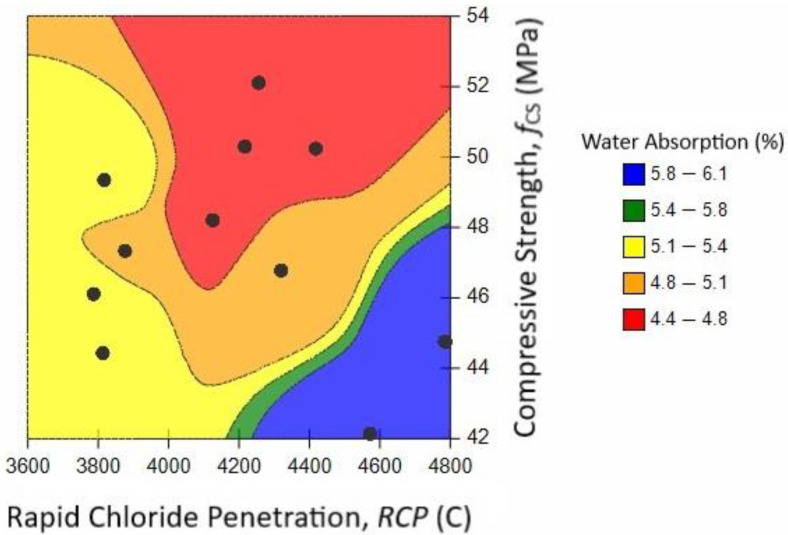
Relationship between RCP, f_CS_, and WA.

**Table 1 materials-14-05933-t001:** Properties of waste mineral admixtures (MS, FA, BFS, and RHA) and cement.

Property	Cement	MS	FA	BFS	RHA
CaO (%)	64.3	0.35	4.2	43.4	1.8
SiO_2_ (%)	20.4	98.3	57.1	36.2	90.5
Al_2_O_3_ (%)	7.1	-	29.7	14.6	0.33
MgO (%)	2.8	-	-	-	0.62
Fe_2_O_3_ (%)	3.4	-	11.9	1	0.41
SO_3_ (%)	1.3	-	0.24	-	
Loss of ignition (800 °C)	1.1	0.8	4.61	3.22	5.41
Specific-surface area (m^2^/kg)	365	27,500	305	341	368
Sp.gravity	3.13	2.22	2.44	2.94	2.14

**Table 2 materials-14-05933-t002:** Aggregates’ properties.

Aggregate Type	Source	Water Absorption (%)	Bulk Density (kg/m^3^)	Specific Gravity
Fine aggregate	Lawrancepur siliceous sand	0.78	1625	2.68
NCA	Crushed limestone	0.67	1547	2.67
RCA	Normal strength concrete samples	5.34	1375	2.37

**Table 3 materials-14-05933-t003:** Nomenclature of mixes.

Mix Nomenclature	Fiber	Mineral Admixture	RCA	Superplasticizer
CON	x	x	x	x
PRAC	x	x	✓	x
PRAC-10MS	x	10% MS	✓	✓
PRAC-30BFS	x	30% BFS	✓	x
PRAC-20FA	x	20% FA	✓	x
PRAC-15RHA	x	15% RHA	✓	x
GFRRAC	1% Microglass fiber	No	✓	✓
GFRRAC-10MS	1% Microglass fiber	10% MS	✓	✓
GFRRAC-30BFS	1% Microglass fiber	30% BFS	✓	✓
GFRRAC-20FA	1% Microglass fiber	20% FA	✓	✓
GFRRAC-15RHA	1% Microglass fiber	15% RHA	✓	✓

✓—Used; x—Not used.

**Table 4 materials-14-05933-t004:** Details of mix compositions.

Mix Nomenclature	Type I Cement (kg/m^3^)	MS (kg/m^3^)	BFS (kg/m^3^)	FA (kg/m^3^)	RHA (kg/m^3^)	NCA (kg/m^3^)	RCA (kg/m^3^)	Fine Aggregate (kg/m^3^)	Water (kg/m^3^)	Fiber (kg/m^3^)	Superplasticizer (kg/m^3^)
CON	565	x	x	x	x	820	x	664	215	0	0
PRAC	565	x	x	x	x	x	720	664	215	0	0
PRAC-10MS	509	42	x	x	x	x	720	664	215	0	2.65
PRAC-30BFS	396	x	159	x	x	x	720	664	215	0	0
PRAC-20FA	452	x	x	88	x	x	720	664	215	0	0
PRAC-15RHA	480	x	x	x	58	x	720	664	215	0	0
GFRRAC	565	x	x	x	x	x	707	651	215	26	3.89
GFRRAC-10MS	509	42	x	x	x	x	707	651	215	26	5.52
GFRRAC-30BFS	396	x	159	x	x	x	707	651	215	26	3.17
GFRRAC-20FA	452	x	x	88	x	x	707	651	215	26	3.46
GFRRAC-15RHA	480	x	x	x	58	x	707	651	215	26	3.95

x—Not used.

## Data Availability

The data presented in this study are available upon request from the corresponding author.

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
