# Peer review of "Evaluation of Mechanical and Permeability Characteristics of Microfiber-Reinforced Recycled Aggregate Concrete with Different Potential Waste Mineral Admixtures"

_materials, 2021, doi:10.3390/ma14205933_

Round 1

Reviewer 1 Report

In this article, the authors have evaluated the mechanical and permeability characteristics of micro-fiber reinforced recycled aggregate concrete with different potential waste mineral admixtures. The detailed comments are as follows:

  1. The introduction section needs revision. Authors should discuss the recent developments to improve the performance of recycled aggregate concrete. Authors may consider https://doi.org/10.1016/j.coldregions.2020.103126. There are many references in the mentioned study, which can improve the quality of the introduction.
  2. All the waste materials (steel slag, coal fly-ash-class F, rice husk-ash and micro-silica) considered in this study are commonly known. Therefore, the details of these wastes in the introduction section are unnecessary.
  3. Authors have referred to a lot of their articles on similar topics. Authors should clearly discuss the difference between their previous articles and this study. It is important to know that the results of this study are original.

Author Response

Dear Reviewer,
Enclosed please see the revised version of the manuscript and responses to your valuable comments.

Reviewer 2 Report

Manuscript ID: materials-1407149

Title: Evaluation of mechanical and permeability characteristics of

micro-fiber reinforced recycled aggregate concrete with different potential

waste mineral admixtures

Authors: Babar Ali, Rawaz Kurda *, Ahmed Salih Mohammed, Rayed Alyousef,

Hisham Alabduljabbar

This paper provides some aspects of the evaluation of mechanical and permeability characteristics of micro-fiber reinforced recycled aggregate concrete with different potential waste mineral admixtures. In general, this work is valuable, nevertheless there is no scientific discussion involved.

I will accept this paper, but some improvements must be made.

Table 1 – oxides should be expressed in wt. %, probably. The unit of loss of ignition is probably also in wt. %.

Water content and fibers should be presented in the amount over 100% of dry components.

Could the Authors provide deep discussion on workability results, especially, chemistry of binders/reaction after adding of water should be clarified.

Could the Authors provide some methods of materials characterization, e.g. the progress of mechanical strength increase should be discussed in the terms of XRD, SEM etc.

I accept this work after correction and answers to my comments from the Authors presented above.

Author Response

Enclosed please see the revised version of the manuscript and response to your valuable comments.

Reviewer 3 Report

  • Abstract: what do you mean by "efficiency of micro-fibers in RAC"? Efficiency in terms of what?
  • State what is novel about this work? what contribution to knowledge does it make compared to what is already known?
  • Include the following very recent references:
    - Extended application of the Equivalent Mortar Volume mix design method for recycled aggregate concrete
    EE Anike, M Saidani, AO Olubanwo, M Tyrer. 5th World Congress on Civil, Structural, and Environmental Engineering (CSEE'20)
    - Effect of mix design methods on the mechanical properties of steel fibre-reinforced concrete prepared with recycled aggregates from precast waste
    EE Anike, M Saidani, AO Olubanwo, M Tyrer, E Ganjian. Structures 27, 664-672
  • "Furthermore, replacement of some parts of cement with 147
    mineral admixtures substantially reduces the carbon footprint of FRRAC":  can you quantify "substantially"? what sort of reduction is achieved in percentage per cubic meter for example?
  • Does the age of the crushed concrete specimens matter? how can it affect the results? comment.
  • Eq.(3): did you assume a linear variation of the axial deflection? explain why, either way.
  • Did you determine the densities of the specimens: wet and dry? explain and discuss.
  • Check English spelling and grammar throughout - I have spotted some errors: for example in section 3.3, last sentence should read "and finally subjected..." (instead of final).

Author Response

(The authors gave the same response as above.)

Round 2

Reviewer 1 Report

Authors have revised the article as per reviewer comments. So, it is suitable for publication.

Author Response

Thank you very much for your valuable comments.